# The Role of Entropy in Construct Specification Equations (CSE) to Improve the Validity of Memory Tests: Extension to Word Lists

**DOI:** 10.3390/e24070934

**Published:** 2022-07-05

**Authors:** Jeanette Melin, Stefan Cano, Agnes Flöel, Laura Göschel, Leslie Pendrill

**Affiliations:** 1Department of Measurement Science and Technology, Research Institutes of Sweden (RISE), AWL Sven Hultins Plats 5, vån 4, 412 58 Göteborg, Sweden; jeanette.melin@ri.se; 2Modus Outcomes, Spirella Building, Letchworth Garden City SG6 4ET, UK; stefan.cano@threadresearch.com; 3Department of Neurology, University Medicine Greifswald, 17475 Greifswald, Germany; agnes.floeel@med.uni-greifswald.de; 4German Center for Neurodegenerative Diseases (DZNE), Standort Rostock/Greifswald, Germany; 5Department of Neurology, Charité—Universitätsmedizin Berlin, Corporate Member of Freie Universität Berlin and Humboldt-Universität zu Berlin, Charitéplatz 1, 10117 Berlin, Germany; laura.goeschel@charite.de; 6NeuroCure Clinical Research Center, Charité—Universitätsmedizin Berlin, Corporate Member of Freie Universität Berlin and Humboldt-Universität zu Berlin, Charitéplatz 1, 10117 Berlin, Germany

**Keywords:** entropy, information, metrology, measurement system analysis, Rasch, cognition, memory, task difficulty, person ability, neurodegenerative diseases, cognitive neuroscience

## Abstract

Metrological methods for word learning list tests can be developed with an information theoretical approach extending earlier simple syntax studies. A classic Brillouin entropy expression is applied to the analysis of the Rey’s Auditory Verbal Learning Test RAVLT (immediate recall), where more ordered tasks—with less entropy—are easier to perform. The findings from three case studies are described, including 225 assessments of the NeuroMET2 cohort of persons spanning a cognitive spectrum from healthy older adults to patients with dementia. In the first study, ordinality in the raw scores is compensated for, and item and person attributes are separated with the Rasch model. In the second, the RAVLT IR task difficulty, including serial position effects (SPE), particularly Primacy and Recency, is adequately explained (Pearson’s correlation R=0.80) with construct specification equations (CSE). The third study suggests multidimensionality is introduced by SPE, as revealed through goodness-of-fit statistics of the Rasch analyses. Loading factors common to two kinds of principal component analyses (PCA) for CSE formulation and goodness-of-fit logistic regressions are identified. More consistent ways of defining and analysing memory task difficulties, including SPE, can maintain the unique metrological properties of the Rasch model and improve the estimates and understanding of a person’s memory abilities on the path towards better-targeted and more fit-for-purpose diagnostics.

## 1. Introduction

When searching for novel diagnostics for cognitive decline, serial position effects (SPE) in memory recall tests have recently come to the fore [1]. A SPE refers to the relationship between the placement of a symbol (e.g., word) in a list and its likelihood of being recalled. For free recall in word learning list tests, SPEs mean that the first words (primacy region, Pr) and the last few words of a list (recency region, Rr) are easier to remember than items in the middle region (Mr) [2]. SPEs in word learning tests, such as Pr and Rr, have been studied since the 1950s [1,2,3,4,5,6,7,8,9].

SPEs were evident in several memory recall studies, as noted in the review of Hurlstone et al. [10]: “*Forward accuracy serial position curves exhibiting effects of primacy and recency are not confined to verbal memoranda. The forward serial position curves associated with the recall of sequences composed of various types of nonverbal stimuli have been shown to exhibit an extensive primacy effect accompanied by a one-item recency effect. These stimuli include visual–spatial locations, visual–spatial movements, auditory–spatial locations, visual matrix patterns, and unfamiliar faces*”.

Of particular interest is to metrologically validate and verify a recently proposed novel diagnostic for cognitive impairment based on the observation that less cognitively able individuals cannot benefit from the SPE of Primacy [1], which, for more cognitively able individuals, makes it easier to recall the first few words in a word learning list test such as Rey’s Auditory Verbal Learning Test RAVLT.

Weitzner & Calamia [1], in their recent review of the diagnostic potential of SPE, concluded that: “*The analysis of SPEs has demonstrated some utility as a marker of cognitive impairment associated with MCI, AD, and other dementias; however, research is limited by the multiple ways in which SPEs are defined and analysed*”.

The overall aim of the present work is to highlight entropy-based explanations when developing metrological methods to word learning list tests, including a better definition and analysis of SPEs, by extending earlier simple syntax studies [11,12]. As recounted in Appendix A, the concept of entropy can be usefully deployed when explaining such effects throughout the measurement process.

In the next section, we introduce the participants and memory test used in this paper. These are applied in three case studies: (I) compensating for the ordinality of raw scores and separating item and person attributes, (II) linking entropy to CSE, and (III) evaluating the multidimensionality introduced by SPE. Each case study on the word recall test RAVLT exemplifies the specific application of a general, entropy-based theory presented in the set of Appendix A, Appendix A and Appendix A.

The paper concludes with a consideration of how the present research contributes to developing a more consistent way of defining and analysing SPE when aiming for better-targeted and more fit-for-purpose diagnostics, as well as introducing the metrological quality assurance of ordinal properties more generally.

## 2. Materials

The present work analyses data from the European EMPIR 19HLT04 NeuroMET2 [13] and SmartAge [14] cohorts, comprising a total of $N_{TP}=$ 225 participants, including *n* = 66 healthy controls (HC), *n* = 99 with subjective decline (SCD), *n* = 27 with mild cognitive impairment (MCI), and *n* = 33 with dementia due to suspected Alzheimer’s disease (AD). Further details about the cohorts can be found elsewhere [13,14]. There are a number of commonly used neuropsychological tests for assessing different cognitive functions, such as memory, learning, speed, attention, visuospatial, language, and executive functions. Measuring memory abilities based on the recall of word list items has a long tradition in neuropsychological assessments and has recently become a research topic for improved diagnostics for cognitive decline and dementia.

One such “legacy memory test” is the RAVLT [15], where a list of 15 semantically unrelated words is orally presented in an initial trial to the test person, who is required to freely recall the words (*immediate recall (IR)*). Subsequently, four more identical trials are performed, making a total of five IR (or learning trials). After these trials, a distractor list of 15 different and unrelated words is presented, and the test person is asked to recall them. This distraction list is directly followed by asking the test person freely to recall again the words from the first presented list, and finally, after a delay of approximately 20 min, they are asked to freely recall the words once again (*delayed recall (DL)*). In addition, there is a list of 50 words, of which the test person is asked to recognise the 15 initial words (*recognition*).

## 3. Case Study I: Analyses of Word-List Task Difficulty

This first case study focuses on the difference between traditional analyses with classic test theory (CTT) and modern metrological analyses—Rasch generalised linear modelling (GLM)—of the observed response data from RAVLT IR, particularly when allowing for SPE when estimating both person memory abilities and task difficulties (Section A.1).

### 3.1. Classic Analyses in Word Learning List Tests

An observed response in a word learning list test is typically scored binarily in one of two categories: as *c* = 1 for correct and *c* = 0 for incorrect, and presented as $P_{success}$. As mentioned in Section A.1, these observed responses constitute the raw data, $x_{i,j}$, for test person *i* and task item *j,* which are characterised by ordinality and are not measures of the person’s memory ability or the memory task difficulty.

CTT, which has been the dominating approach in previous analyses of word learning tests reported in the literature [1], simply sums up the number of correct recalled words; therefore, for instance, an average score for task item *j* is: $CTT_{proportion,j}=\frac{1}{N_{TP}}\cdot \sum_{i=1}^{N_{TP}}x_{i,j}$, where $N_{TP}$ is the number of test persons in the cohort.

### 3.2. Modern Analyses in Word Learning List Tests

Compensating for ordinality and providing separate estimates of task difficulty, δ, and person ability, θ, (so-called specific objectivity) is done in modern analyses by making a logistic regression, based on Equation (A2), with the Rasch GLM [16] of the log-odds ratio to the raw binary response data, $P_{success,i,j}$ for person *i* and item *j*:(1)$$ln\left(\frac{P_{success,i,j}}{1-P_{success,i,j}}\right)=\left(\theta_{i}-\delta_{j}\right)$$

Such a logistic regression, providing a restituted estimate of the object variable Z from the measured response Y (Equation (A1)), can be readily done with, for example, the WINSTEPS^®^ software application (version 3.80.1 is used here).

Reflecting the nonlinearity and ordinality due to the counted fractions mentioned in Section A.1, typically, S- or ogive curves (Figure 1) are expected from Equation (1) when correlating the observed responses $P_{success}$ with the test item difficulty, δ, and person ability, θ. The increasing nonlinearity of the curves as one approaches each end of the scales is apparent in Figure 1, despite the large uncertainties. The straight line plotted in Figure 1a indicates the slope of the Rasch formula (Equation (1)) at its maximum value—that is, at $P_{success}=50\%$, where $\delta =\bar{\theta }=-0.58\;logits$, the average person’s ability across the cohort. From Figure 1a, task difficulties at the low end of the scale (the easiest tasks) are visibly underestimated, while, at the high end, the more challenging tasks are overestimated with CTT.

A well-designed test with good targeting, where the span of a person’s abilities matches well (both in terms of centring and width) the corresponding range of task difficulties, will admittedly, in general, have small nonlinearities, as is the case for the present work with RAVLT IR. These effects nevertheless have to be investigated, however large or small they might turn out to be. Note that, in any case, the Rasch model is essential for metrological quality assurance, since it provides, through measurand restitution, separate values for task difficulty and person ability.

The classic diagnostics of SPEs, such as Pr and Rr, are traditionally plotted as curves (Figure 2a) that attempt to visualise the trends in the raw scores for the test items between the three main regions. Typically, in a word learning test with 15 words: *k* = 1, …, 15, such as the RAVLT, Pr is normally found to be significant only for the first four words: *k* = 1,…, 4, while Rr is found in the last four words: *k* = 12,…, 15 of the recalled list [7]. Traditional calculations using CTT for SPE, as reported by Talamonti et al. [9], consist of simply summing up the number of correct recalls in the Pr and Rr, respectively. For comparison, we applied both methods—CTT and Rasch—to our data set of the NeuroMET cohort.

Figure 2a shows the proportion, *P_success_*, of immediately recalled (trial 1) words for Pr, Mr, and Rr based on the CTT raw scores, while Figure 2b shows corresponding analyses from Rasch person attributes converted back to $P_{success}$, where the latter includes compensation for the scale nonlinearities shown in Figure 1. Although ordered according to the expected cognitive health status of each clinical subgroup, when considering the actual measurement uncertainties (shown in Figure 2b), there appear to be no detectable differences between the four clinical subgroups (HC, SCD, MCI, and AD). Note that we refrain from plotting uncertainty intervals on the CTT (Figure 2a), since—without an honest assessment of the effects of ordinality (which we show to be significant in Figure 1a)—it is difficult to give uncertainty estimates objectively and credibly. When uncertainty intervals are plotted in published CTT versions such as Figure 2a, those intervals are often shown (e.g., [5]) incorrectly as symmetric about the mean CTT raw score, which is inconsistent with the inherent nonlinearity of the *y*-axis due to the well-known counted fractions effect, as evident in the asymmetric uncertainty intervals shown in Figure 2b.

As noted above in relation to Figure 1a, because of the counted fractions effect (Section A.1), the task difficulties will be increasingly overestimated with CTT (Figure 2a) as one approaches the low end of the scale (the easiest tasks, such as Pr and Rr). In particular, the apparent prominence in traditional CTT plots (Figure 2a) of the SPE in the Rr, especially for the AD cohort (yellow line), is not maintained when allowance is made for the scale nonlinearity and uncertainties (Figure 2b).

### 3.3. Potential, Limitations, and Implications with a Corrected Analysis of SPEs in RAVLT

This case study has provided both a theoretical justification and experimental demonstration for a proper analysis of RAVLT with a modern metrological approach, the Rasch GLM [16]. With a correct analysis of the SPEs in RAVLT, we thereby move away from the limitations of traditional CTT and the associated lack of metrological invariance and considerable risks of incorrect conclusions based on the cognitive assessment data, which would otherwise impact both clinical decision-making and the assessment of the significance of correlations between cognitive functioning and brain structure and biochemistry [17].

In fact, our present analysis indicates, by considering the actual measurement reliability, that there is hardly any detectable difference between the four clinical subgroups, and the diagnostic power of the SPEs is rather weak, owing to the relatively large measurement uncertainties, even when a proper analysis of the raw scores has been made. Therefore, our main message here is to define a methodology for dealing properly with ordinal data.

To deal with cases where appreciable portions of the cohort, for some reason or other, respond differently to particular items—e.g., those items prone to SPEs such as recency and primacy—tests to confirm the unique metrological properties of the Rasch model [16] will be examined in Section 5. However, before that, in the next section, we consider how SPEs can be explained, amongst others with a causal, entropy-based theory and in support of validity.

## 4. Case Study II: Explaining Serial Position Effects

In case study I (Section 3 above), our experimental observations were used to obtain empirical estimates of the memory task difficulty and person memory ability (as plotted on the *x*-axes of Figure 1a,b, respectively, and given below in Table 1) by a methodological analysis of the experimental responses, with a logistic regression based on the Rasch formula [16] in Equation (1).

Here, we complement these analyses with ab initio theoretical and quasi-theoretical causal estimates of the memory task difficulty, thus providing evidence for construct validity, item equivalence, and enabling metrological references based on causality and best understanding. Our previous work on recalling the simplest nonverbal items—taps and numbers [11,12]—gave first principle, ab initio explanations of the task difficulty in terms of entropy: less entropy means a more ordered task, which is thus easier to perform. This entropy-based explanatory theory, summarized in Appendix B, is extended here to include causal explanations of recall in the word learning list test RAVLT IR, including SPE such as primacy and recency.

Based on this entropy approach, we introduce CSE as a quasi-theoretical method to explain the task difficulty for word recall in general (Section B.2) and SPE in particular (Section B.4).

### 4.1. Construct Specification Equations (CSE) to Explain Task Difficulty in Word Learning Tests

Expressing a CSE for task difficulty in the form of Equation (A3) can be done in terms of the total entropy as a prediction of the task difficulty for all the symbols (words), *j*. The total task difficulty will then be simply found by adding the separate entropies discussed in Appendix B (Equations (A5)–(A8)):(2)$$\delta_{j}=\delta_{j,Pr}+2\cdot \delta_{j,Mr}+\delta_{j,Rr}+\delta_{j,freq}$$where $M=\frac{1}{ln\left(L\right)}=\frac{1}{ln\left(15\right)}=0.369$ for a sequence of length, and L=15 words.

Equation (2) is an example of a pure ab initio theoretical definition of a task difficulty in RAVLT IR, where there are potentially four separate components: the mid-range, primacy, and recency word recall difficulties, plus word frequency. It is not obvious that the chosen explanatory variables in Equation (2) will be the principal components of the variations in the empirical data. The various coefficients $\beta_{k}$, as they occur in Equations (A3) and (A5)–(A8), need to be determined. A PCR analysis (Section B.1) was therefore performed. As a result of the initial PCA, it was found that the primacy and recency explanatory variables were correlated with a Pearson’s coefficient of −0.94 (where the minus sign indicates that an increase in primacy is accompanied by a corresponding decrease in recency and vice versa). Bearing in mind the similarities between the theoretical expressions of Equations (A6) and (A7), such a high correlation is to be expected. On the other hand, there was only a weak correlation (coefficient −0.18 at most) found from the PCA between recency or primacy (Section B.4) and word frequency (Section B.3).

Based on the empirical task difficulty parameters from the present data set and the four explanatory variables (Equations (A5)–(A8)), the resulting CSE from a PCR analysis of the task difficulty of item *j* in RAVLT IR for the whole cohort is found to be (uncertainties, with coverage factor *k* = 2, are given in parentheses):(3)$$zR_{RAVLT\;IR,j}=+5\left(3\right)+0.7\left(5\right)\cdot {Primacy}_{j}+0.8\left(5\right)\cdot {Recency}_{j}+0.2\left(2\right)\cdot Frequency_{j}$$
as the restituted value (R) of the object attribute Z, where the explanatory variables X are: $Primacy_{j}=\delta_{j,Pr}$; $Recency_{j}=\delta_{j,Rr}$; and $Frequency_{j}=\delta_{j,freq}$. The CSE such as Equation (4) indicate the predominance of entropy-based syntax contributions in recall task difficulty, while additional semantics effects, such as word frequency, are found to be small and dominated by measurement uncertainties.

Table 1 provides empirical (*δ*, Equation (1)) and CSE/quasi-theoretical task difficulty (zR, Equation (3)) values, as well as the explanatory variables for each RAVLT IR item, and Figure 3 shows correlation plots between the empirical and quasi-theoretical task difficulty values. As an important measure of the goodness of fit, the Pearson’s correlation coefficient, *R*, between the empirical task difficulty values (*δ*) and CSE/quasi-theoretical (zR) task difficulty values was found to be 0.80, which we suggest indicates a satisfactory prediction of the task difficulty.

The contribution from each of the explanatory variables in Equation (3) is shown graphically in Figure 4. An inspection of that plot, as well as those in Figure 5a–c, confirms the observations of the earlier researchers, who characterised Pr as a SPE of the first four words of the list, whilst Rr was a SPE of the last four words [9,18,19,20]. Note, however, that according to our entropy-based theory (Section B.4), the relative reductions in task difficulty of Pr and Rr (red and green curves in Figure 4) vary greatly across all item orders but complement each other (a large Pr is accompanied by a small Rr and vice versa, in agreement with the strong, negative correlation between these found in the PCR). The sum of these negative contributions at any one item *j* is substantially balanced by the average positive contribution from the mid-list term, $\delta_{j,midrange}$, as in Equation (2). (The mid-list term, the intercept, will be discussed further in the third case study, Section 5).

The current work expands on the results of previous studies by being able to explain the various components of the task difficulty in terms of informational entropy, providing satisfactory descriptions of, in particular, SPE such as primacy and recency, simply in terms of combinatorics and Brillouin’s [21] entropy expressions (Equations (A5)–(A8)).

Our theoretical explanation of the SPE task difficulty in terms of entropy is considerably simpler than earlier theories of SPEs, such as reviewed by Hurlstone [10], which invoked additional explanations in terms of memory processes, including (i) a primacy gradient of the activation levels (i.e., first item activated the strongest, and a monotonically decrease for the latter items), (ii) a cumulative match (i.e., matching of incoming sequences to previously presented sequences), (iii) a revised primacy model (i.e., a sequence is made familiar through repetition in the local chunk nodes), and (iv) an output inference model (i.e., adding increasing amounts of Gaussian noise with each successive one). Such additional effects would need to be studied further in extended investigations, hopefully with smaller measurement uncertainties than in the present work.

### 4.2. Construct Specification Equations for Cohort Subgroups

Having explained the task difficulty, the next step is to investigate how well RAVLT IR and, in particular, the SPE can distinguish between different cognitive health statuses in view of the purported new diagnostic tool (Section 1). With the same explanatory variables, but with different empirical task difficulties, the values for the two principal groups of the test persons portioned according to cognitive health status are found to result in slightly different CSEs for each group, although these differences are barely significant compared with the measurement uncertainties (the latter, with a coverage factor *k* = 2, are given in parentheses):(4)$$zR_{RAVLT\;IR\;HC+SCD}=+4\left(3\right)+0.7\left(6\right)\cdot Primacy{\;}_{i}+0.7\left(5\right)\cdot Recency{\;}_{i}+0.2\left(2\right)\;\cdot Frequency_{i}$$
(5)$$zR_{RAVLT\;IR\;MCI+AD\;}=+7\left(4\right)+1.0\left(7\right)\cdot Primacy{\;}_{i}+0.9\left(7\right)\cdot Recency{\;}_{i}+0.1\left(3\right)\;\cdot Frequency_{i}$$

Figure 5a–c show the relation between the empirical values and CSE/quasi-theoretical (zR) values of the task difficulty for the whole cohort and the two subgroups.

As observed in the earlier literature, cognitively impaired individuals are not always able to benefit from reductions in the recall difficulty normally facilitated by SPE, particularly Pr. This health status increase in the recall task difficulty for Pr items is the basis for claiming SPE as a purported diagnostic tool for cognitive decline (Section 1). As shown in Figure 5c, these claims are confirmed here by the reduction in Pr recall task difficulty, which we find to be less pronounced for the least cognitively healthy cohort group, while the Rr effects appear largely unchanged.

When making this interpretation, it is worth recalling that a reduction in the number of test persons—as in the analyses in these cognitive healthy-specific groups—leads to increases in the measurement uncertainties, as also reflected in the CSE, in Equations (4) and (5) compared to Equation (3). Uncertainties in the *β*-coefficients in each CSE are a combination of the measurement uncertainties, *U*δ, in the empirical task difficulty values (typically 0.3 logit, as shown in Figure 5a–c), which propagate through principal component regression together with the modelling uncertainties (PTC MATHCAD Prime 6.0.0.0, module *polyfitc*), which are an order of magnitude larger and arise, for example, if additional components or other sources contribute to the unexplained variance. A further point to be made on the interpretation and its relation to the number of test persons: The cognitively healthier group (HC and SCD) comprises 165 persons, while the less cognitively healthy group (MCI and AD) only comprises 60 persons. This uneven balance is reflected in the very similar Equations (4) and (5) and Figure 5a,b compared to Equation (6) and Figure 5c.

Despite the large uncertainties, reasonable agreement is found between the theory and experiment, where the Pearson’s correlation coefficient *R*, as a measure of the agreement between the empirical task difficulty values and CSE/quasi-theoretical (zR) task difficulty values, is found to be slightly larger for the group of MCI and AD, *R* = 0.85, compared to HC and SCD, *R* = 0.75.

### 4.3. Insights into Serial Position Effects Based in Word Learning List Tests on Construct

This second case study started by introducing the general role of the CSE as an advanced method for testing theories of constructs, particularly task difficulty, provided by an analysis of the empirical data using the modern metrological approach, the Rasch GLM model, from the first case study. This second case study concluded with an enhanced understanding of the SPE explained in entropy terms for RAVLT IR and presented a new methodology for calculating the recall task difficulty in word learning list tests.

To identify an improved diagnostic tool for cognitive decline based on the SPE, it is important to consider that SPE are prominent contributions to task difficulties, which vary from word to word more rapidly than the average difficulty of recalling an arbitrary word anywhere on the list, as explained with the CSE.

For SPE to act as a valid diagnostic tool to identify and predict cognitive impairment, there has to be a causal effect—that is, a significant change in the metric that is related in some way to the cognitive health status of the test person. The diagnostic effects also have to be significant—that is, larger than the measurement uncertainties, where the latter have to be evaluated.

Further studies remain to be done about how the diagnostic sensitivity correlates with the cognitive health of each cohort member, which may be expressed in terms of the explanatory variables (such as biomarkers) in the CSE for a person’s (instrument) ability [22,23]. Another approach—pursued in the final case study (below)—will address the evidence of (perceived) changes ∆δ in the average task difficulty for different parts of the cohort grouped clinically according to the cognitive status and interpreted in terms of the apparent scale distortions, multidimensionality, and resultant changes (residuals) in response.

## 5. Case Study III: Maintaining the Unique Metrological Properties of the Rasch Model in the Presence of Serial Position Effects

Despite the promising new diagnostic tool based on SPE, a caveat is that the unique metrological properties of the Rasch model [16] need to be maintained, particularly in the presence of SPEs, so that the counted fraction ordinality in the raw response scores can be adequately compensated for and unidimensional, separate estimates can be made for each item difficulty (irrespective of the test person) and each person ability (irrespective of the item), as mentioned at the end of case study I (Section 3.3).

The separation of the conjoint Rasch attributes—task difficulty and person ability—necessary for the metrological quality assurance of any indirect ordinal response is usually done by exploiting the property of the specific objectivity of the Rasch model [16] (Equation (1)) (Section A.1). Analysing data sets as a candidate diagnostic tool, including the SPE in word learning list tests (Section 1), can present challenges. If different cohort members experience SPE differently—for instance, according to the cognitive status (as claimed for the new diagnostic tool)—then the task difficulty might include a person-dependent factor, thereby breaking the specific objectivity. This final case study examines the tests to maintain the unique metrological properties of the Rasch model [16].

### 5.1. Scale Distortion and Instrument (Test Person) Discrimination in the Presence of SPE

There is some evidence in our results (Section 4.2) that the benefit from Pr in reducing the task difficulty is particularly diminished for the less cognitive healthy part of the cohort, apparently in line with earlier CTT-based claims about the diagnostic potential of SPE, where patients with AD do not recall the last-mentioned words. Despite large measurement uncertainties, there also appear to be some differences in the first “intercept” term on the RHS of the CSE for task difficulty—Equations (4) and (5) derived from PCR—between the two cohort groups, according to the cognitive status.

In this work, support is taken from our entropy-based theory: A theoretical explanation of the intercept in the CSE for the task difficulty for item *j* is that it equals twice the mid-range task difficulty:$$\delta_{j,Mr}=+M\cdot ln\left(\frac{L}{2}!\right)$$
according to Equation (2), as explained in Section 4.1. An evaluation of that term for a RAVLT list of 15 words yields $\delta_{j,Mr}=+6.5\;logits$, which is close to the experimentally observed intercept, i.e., +7(4) logits in the CSE for the less healthy cohort (Equation (4)). The lower intercept value for the healthy group (HC and SCD), i.e., +4(3) logits (Equation (5)), means that, apart from benefitting from the SPE, the group also has an overall reduced task difficulty compared with the less cognitively healthy group (MCI and AD) across the tasks. SPE, which are appreciable for the healthy group, thus cause an extra discrepancy $∆\delta_{SPE}$ of about 2 logits compared with our entropy-based theory (which is applied better in the less cognitively healthy group, since they lack SPE). Our intercept term values are comparable with the corresponding values of $Asymptote=\delta_{j=\frac{L}{2}}$ reported in earlier studies of Murdock for a wide range of word lists (*L* = 10,…, 40) [2].

A change in the intercept value could be linked to an entropy-based change in the effective length, *L*, of the sequence to be recalled: a change of −2 logits between Equations (4) and (5) would correspond to an effective shortening for the cognitively healthy group (HC and SCD) of the sequence length from 15 to 10 words. For a less healthy group (MCI and AD), the effective length of the list remains 15 words. Similar effects have been found recently, where effective shortening was interpreted as an effect of learning when repeated trials of the same word list were made (as examined in detail in a separate publication [24]).

### 5.2. Principal Components and Multidimensionality

Loading plots of fit residuals such as shown in Figure 6 can reveal evidence—such as the clustering into two SPE groups—for multidimensionality in the principal component analysis (PCA2) of the item logistic regression fit residuals (Section C.2). Such multidimensionality might arise from the SPE, since they imply that each cohort individual might potentially “assess” the task difficulty in an individual way, thus breaking the conditions for what Rasch [16] called “specific objectivity” (Section C.1) needed to make the necessary separation of the task difficulty from the person’s ability.

The present work argues that, because we can explain a task difficulty when formulating the CSE, particularly deploying the concept of entropy (Equation (2)), it may also be possible to identify factors common to both the CSE PCA (as the first step in a PCR, Section 4.1) and the PCA of fit residuals (Figure 6). The effects of scale distortion should therefore be predictable—for instance, in loading plots, as in the present study, where different test persons have more or less discrimination in SPEs, according to their cognitive state, as is now examined empirically.

#### 5.2.1. PCA1 for CSE Formulation

The first kind of PCA (Section C.1), **PCA1**, made as the first step in PCR when forming the CSE (Section 4.1), deals directly with the Rasch parameters, such as task difficulty, *δ*, in terms of a set of explanatory variables, $X_{k}$:$$PC_{p}=\overset{K}{\underset{k=1}{\sum }}a_{p,k}\cdot X_{k}$$

PCA1 is expected to reveal additional components of variation (explanatory variables, $X_{k}$), for example, from the SPE when the CSE (Equation (2)) are formed for the construct task difficulty. The amount of scale distortion arising from the primacy and recency are simply fractions of the task difficulty (given theoretically term-for-term in CSE Equation (2) for each dimension in relation to the overall task difficulty).

From the PCA1 of the current empirical Rasch data, the dominant (most variance) principal component (the first column of the matrix *P*) is found to be:(6)$$PC_{1}=\overset{3}{\underset{k=1}{\sum }}L_{k}\cdot X_{k}=+0.77\cdot Primacy-0.64\cdot Recency+0.02\cdot Freq$$
where $L_{k}$ is the loading of the *k*th explanatory variable, $X_{k}$, and where the SPE (primacy, recency, etc.) are estimated theoretically (with Equation (2)). The coefficients $a_{1,primacy}=+0.77$ and $a_{1,recency}=-0.64$ in Equation (6) will turn out to be of particular interest. The next principal components of variation are found to be:$$PC_{2}=-0.51\cdot Primacy-0.60\cdot Recency+0.62\cdot Freq$$
$$PC_{3}=-0.39\cdot Primacy-0.49\cdot Recency-0.79\cdot Freq$$

Each PC in the present case is thus found experimentally to be related in turn to each of the three main explanatory variables for task difficulty, i.e., primacy, recency, and word frequency (“*Freq*”) (respectively expressed by Equation (3)) as indicated in boldface in the three equations above. The CSE for the task difficulty are formed with PCR (Section 4.1) by regressing the Rasch estimates of *δ* against these three principal components of variation.

#### 5.2.2. PCA2 Rasch Logistic Regression Residuals

In the present observations, empirical evidence was found in **PCA2** (Section C.2) that the SPE represented additional dimensions of the task difficulty over and above the difficulty in recalling any word in the RAVLT immediate recall, as can be seen by the clustering in the loading plot (Figure 6a).

However, the evidence was not completely clear-cut: the eigenvalue of the unexplained variance in the first contrast was <2.0 (i.e., supporting unidimensionality), while the disattenuated correlation between cluster 1 (only items from Rr) and cluster 3 (items from both Pr and Mr) was −0.60 (i.e., indicating multidimensionality). Based on Linacre’s [25] further discussion about multidimensionality and PCA2 analysis:“Compare the raw variance explained by the items (present case 18%) with the unexplained variance in the first contrast (present case 9%); is this ratio big enough to be a concern?” In our analysis of the data, the variance of the first Rasch dimension is about double that of the secondary dimension, but the latter is clearly noticeable.“Is the secondary dimension bigger by chance?” In the present case, with the eigenvalue = 2, this is the strength of approximately two items. It is not expected to have a value of more than two items by chance [26], and at least two items are needed to think of the situation as a “dimension” and not merely an idiosyncratic item, according to Linacre.“Does the secondary dimension have substance?” Looking at the loading plot in Figure 6a, the two SPE clusters of items—primacy and recency—are clearly separated vertically (the important direction) from the other (mid-list) cluster of items. Each cluster can be regarded to be important enough and different enough to be explained in terms of separate dimensions, as motivated in the Introduction.One approach to handling the multidimensionality revealed by the PCA2 analyses is to make a separate fit of the Rasch formula to each cluster of items associated with a different dimension. The significance of the differences found in the test person’s ability between logistic regression fits to the different item clusters can be assessed, for example, with statistical *t*-tests [27,28]. Since, in the present case, SPE such as primacy and recency only affect a few words at the start and end of a word list, this means that a separate Rasch analysis for each such small cluster would result in poor reliability in the test person’s ability estimates owing to the reduced numbers of the degrees of freedom [29]. Similarly, analyses of the portions of the whole cohort grouped by health status would also have poorer reliability. Statistical significance tests or correlation plots between sets of a person’s measures from separate Rasch analyses for each suspected dimension or each cohort group according to the cognitive status were therefore judged to not be useful in view of the relatively large uncertainties of the present data.

#### 5.2.3. Combining the Two Kinds of PCA

In addition to the empirical evidence described above, the entropy-based theoretical modelling of the clusters in the **PCA2** loading plots (such as shown in Figure 6) can provide extra support for the validity for SPE multidimensionality, as follows. A change in the task difficulty ∆δ for each item *j* associated with SPE (Section 5.1) can lead to a change in the logistic fit item residual, $y_{i,j}$, of the regression of Equation (1) to raw response data where the task sensitivity $K_{\delta }=\frac{\partial P_{success}}{\partial \delta }$:(7)$${y^{\prime }}_{j,\delta }=y_{j}-\frac{\partial P_{success}}{\partial \delta }\cdot ∆\delta$$
by a simple partial differentiation (Appendix C).

**PCA2**, based on the regression of fit residuals (Section C.2), can therefore contain evidence of SPE-related effects. In particular, from Equation (7), the loading of a PC in the logistic regression residuals will be proportional to the product of the sensitivity ($K_{\delta }$) and a change in perceived task difficulty, ∆δ:(8)$$\mathrm{PCA}2\;\mathrm{loading}:\;L_{p,x}\propto a_{p,x}\cdot \frac{\partial P_{success}}{\partial \delta }\cdot ∆\delta$$

Term-for-term on the RHS of Equation (8):
The loading coefficient $a_{p,x}$ should be the same as deduced in the **PCA1** above (Section 5.2.1) when forming the CSE for task difficulty in terms of the entropy-based explanatory variables (Equation (6)).$K_{\delta }$, the peculiar sensitivity of the instrument (person) in the Rasch model, will “modify” the PCA loading plots correspondingly but can be calculated from a simple differentiation of the dichotomous Rasch formula (Equation (1)) [30] to yield Equation (9). The sensitivity according to Equation (10) has a “resonance-like” behaviour, with a peak in *K* occurring at $\delta =\bar{\theta }$ (the 50% point $P_{success}$), and the sensitivity will fall off symmetrically on either side to approach zero at each end of the task difficulty scale [30].
(9)$$K_{\delta }=\frac{\partial P_{success}}{\partial \delta }=-\frac{e^{\left(\theta -\delta \right)}}{{\left(1+e^{\left(\theta -\delta \right)}\right)}^{2}}$$The third term on the RHS of Equation (9) is any significant change, ∆δ, in the task difficulty. As found experimentally (Section 5.1), the task difficulty of each item deviates the most from the basic Rasch model for the healthy cohort, both owing to the SPE—such as Primacy and Recency—as well as an overall reduction in the task difficulties across all the items: ∆δ=−2 logit. (In contrast, members of the less cognitively healthy portion of the cohort benefit less from the Primacy and have a mid-range task difficulty close to the theoretical value of the CSE intercept).


The final step in evaluating the SPE loading according to Equation [8] and accounting for the loading plot of Figure 6 is to integrate the task difficulty sensitivity $K_{\delta }=\frac{\partial P_{success}}{\partial \delta }$ according to Equation (9) to allow for variations in the sensitivity at each level of task difficulty as a function of a person’s ability across the distribution of the cohort. The modified expression is:(10)$$\mathrm{PCA}2\;\mathrm{loading}:\;L_{p,x}\propto a_{p,x}\cdot ∆\delta \cdot \int p\left(\theta \right)\cdot \frac{\partial P_{success}}{\partial \delta }\cdot d\theta$$
where a person’s ability is taken to be normally distributed across the cohort “all” according to:(11)$$p\left(\theta \right)=\frac{1}{\sigma \cdot \sqrt{2\cdot \pi }}\cdot e^{-\frac{1}{2}\cdot \frac{{\left(\theta -{\bar{\theta }}_{all}\right)}^{2}}{\sigma^{2}}}$$
where ${\bar{\theta }}_{all}=-0.6\left(1.0\right)$, the average cohort ability, and σ=1.0, based on our Rasch analysis of the data (Section 3.2).

An inspection of the Rasch residual PCA2 loading plot (Figure 6 for the whole cohort) reveals evidence of these deviations in task difficulty across the items, as investigated for the empirical data for task difficulty plotted in Figure 5a–c and in terms of our entropy-based theory of task difficulty, leading to Equation (2).

The agreement between the experimental and theoretical values of SPE loading evident in Figure 6 is deemed satisfactory, lending support to the empirical evidence for multidimensionality associated with the SPE in the word learning list tests.

## 6. Conclusions

The use of entropy as a powerful explanatory variable was found to explain the difficulty of recalling word sequences as an extension of an earlier work on syntax-based and nonverbal memory tests as part of our research to provide descriptions of the intrinsic properties of a measurement system, where a test person acts as a measurement instrument.

Brillouin’s classic combinatoric entropy expression (Equation (A4)) explains both the overall task difficulty, as in our earlier studies of nonverbal tests, as well as the SPE such as primacy and recency in the first trial of the word learning test RAVLT. The CSE, which explain the recall task difficulty, provide evidence for the construct validity, item equivalence, and enable metrological references based on the causality and best understanding. These results complement the conclusions of several previous studies—reviewed by Weitzner & Calamia [1]—which claimed SPEs to be significant diagnostic tools, while, at the same time, not explicitly declaring their measurement uncertainties or using the Rasch model. In response to claims in the literature that *research (into a new SPE-based diagnostic) is limited by the multiple ways in which SPEs are defined and analysed* [1], the present work proposes a more consistent way of defining, analysing, and theoretically predicting SPE multidimensionality while maintaining the unique metrological properties of the Rasch model.

Alternatives to the Rasch model, such as the so-called 2PL and 3PL item response theory (IRT) models, have claimed to handle effects such as discrimination and a number of causes of *measurement disturbances, such as start-up, guessing, plodding, carelessness, and item/person interactions* [31], each of which may plausibly depend on the state of health of each cohort member. However, such alternative IRT do not have the specific objectivity property of the Rasch model essential for its unique metrological properties. According to Wright & Stone [32], *Mathematical analysis shows that the 3PL model is a non-converging, inestimable elaboration of the Rasch model* and has traditionally has been judged to be too idiosyncratic to be useful in a general measurement system [32]. In educational contexts, one can argue, for example, that, instead of reanalysing the responses of a whole class, the one or two individuals showing another discrimination can better be dealt with separately. Such exceptions are, however, not possible in the present healthcare example, since significant portions of the cohort appear to have different levels of discrimination, depending on their cognitive status.

Multidimensionality always exists to some extent [25,33]; the question is whether it is significant enough to warrant separating groups of items into subtests (e.g., in the present case for primacy and recency) in an attempt to maintain the unique metrological properties of the Rasch model [16]. We adopted Linacre’s [34] recommendation, described in Section 5.2.2, to examine the multidimensionality through studies of the residuals of fit for both the raw data and the Rasch attributes.

Our PCA-based approach has some similarities to what has been called scale alignment; in the words of Feuerstahler and Wilson [35], “*[by] projecting the dimensions from the multidimensional model onto a single reference dimension…which represents a composite of the individual dimensions…. scale alignment aims to better achieve the ideal relationship between sufficient statistics and item parameter estimates*”. We consider SPEs to introduce a kind of *between-items multidimensionality*, as studied earlier, e.g., educational studies—[35,36] mention *algebra*, *geometry*, *statistics*, *and written mathematics*—but, in our case, with recall tasks arguably conceptually simpler and capable of being modelled theoretically from the first (entropy-based) principles in support of validity.

Two distinct but related PCAs are examined in detail in Section 5.2, where our entropy-based theory has provided support for the validity of the assessments of multidimensionality associated with the SPE. Our studies show that the effects of scale distortion are predictable to some extent, for instance, in loading plots, where different test persons have more or less discrimination in the SPEs according to their cognitive state.

Our new methodology applies modern measurement theory but prepares us to deal with situations where SPE were to be adopted as a diagnostic tool. To date, in our study, measurement uncertainties have been relatively large, reflecting the limited sample size, collinearity, and measurement disturbances (Section 4.1), and there are sources of dispersion when making multivariate regression that are not yet accounted for. What appears to be the case is that, over and above individual variations in a person’s ability, there is an overall shift in the person’s ability for each clinical group. Whether one regards that as a change in ability or a change in task difficulty is a moot point. Choosing the task difficulty as a metrological reference (as proposed in terms of robustness and simplicity [37,38]) would suggest that SPE dependency would be assigned to a person’s ability.

The next steps include acquiring larger samples in order to explore the multidimensionality effects beyond the measurement uncertainties, other test sequences such as different lengths, and trials, as well as evaluations with other diagnoses. This will both enhance the understanding of the multidimensionality in tests where the SPE may affect the discrimination, as well as provide clinicians with more valid and reliable tools for the diagnosis of cognitive impairment.

Here, we did not focus on the cognitive function or brain structure and networks of each test person (which are more related to explaining a person’s ability, *θ*) but, rather, on explaining the difficulty, *δ*, of the task of recalling each word list, which are the objects of the measurement system. Other parts of the NeuroMET2 project are currently studying correlations between a person’s ability and a number of imaging and liquid biomarkers. Forthcoming works are expected to address the brain structure and networks, also with an emphasis on the concept of entropy for cognitive processes.

From a broader perspective, the present work is part of an ongoing effort aimed at extending traditional metrological concepts to include ordinal properties in response to a widening and increasingly important set of scientific studies and the applications of these properties (such as in education, sustainability, and healthcare).

## Figures and Tables

**Figure 1 entropy-24-00934-f001:**
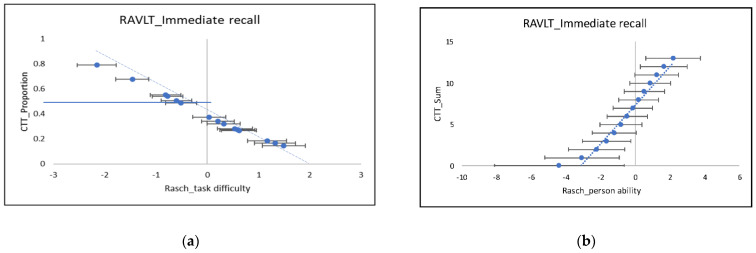
Correlation plots between classic scoring (*y*-axis; CTT proportion in (**a**) and CTT sum score in (**b**) of our actual data for RAVLT IR and corresponding Rasch restituted values (Equation (1)): (**a**) task difficulty, δ, (*x*-axis), where a lower value indicates a lower task difficulty, and (**b**) person ability, θ, (*x*-axis), where a lower value indicates a higher ability. Uncertainty coverage, *k* = 2.

**Figure 2 entropy-24-00934-f002:**
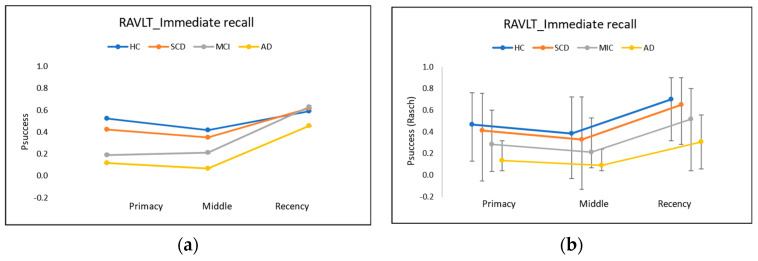
The proportion, $P_{success}$, of recalled words in RAVLT IR (trial 1) for Pr, Mr, and Rr based on the raw scores (**a**) and Rasch attributes converted back to the $P_{success}$ scale in (**b**). AD = Alzheimer’s disease; HC = healthy controls; IR = immediate recall; MCI = mild cognitive impairment; SCD = subjective cognitive decline. Uncertainty coverage, *k* = 2.

**Figure 3 entropy-24-00934-f003:**
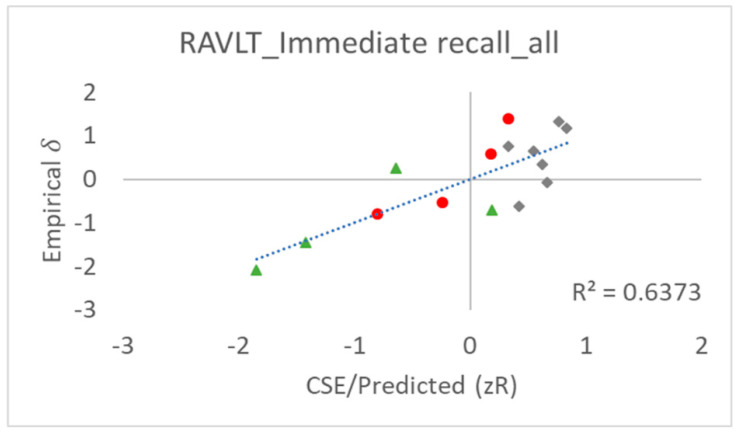
Correlation plot between the empirical task difficulty values (*δ*, Equation (1), *y*-axis) and CSE/quasi-theoretical (zR) task difficulty values (Equation (10), *x*-axis). Item colouring and shape correspond to red dots = Pr items, grey diamonds = Mr items, and green triangles = Rr items.

**Figure 4 entropy-24-00934-f004:**
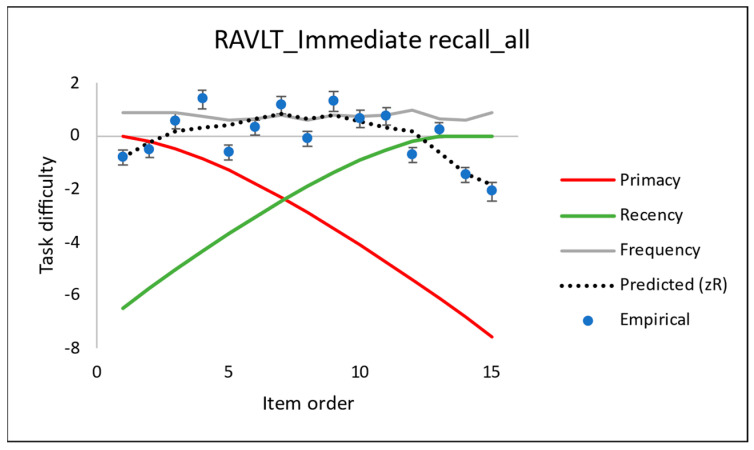
Contribution of the different explanatory variables for the CSE for the task difficulty in RAVLT IR from Equation (4). Uncertainty coverage factor *k* = 2.

**Figure 5 entropy-24-00934-f005:**
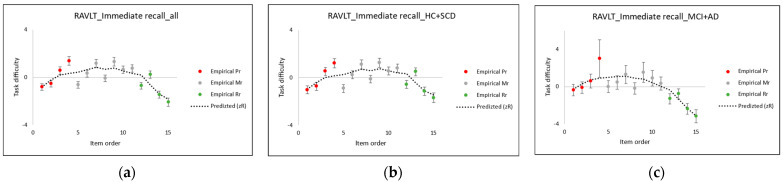
Relations between the empirical values and CSE/predicted (zR) values of the task difficulty for (**a**) the whole cohort and (**b**,**c**) the two subgroups of the cognitively healthy (HD and SCD) and cognitively impaired (MCI and AD). Item colouring corresponds to red = Pr items, grey = Mr items, and green = Rr items.

**Figure 6 entropy-24-00934-f006:**
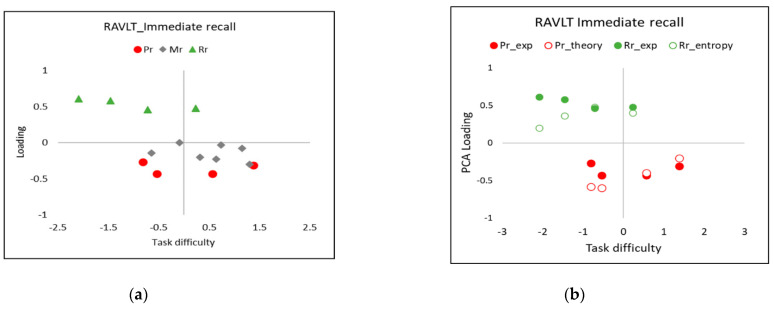
PCA2 loadings (*y*-axes) vs. the RAVLT immediate recall task difficulty, *δ*, of the Rasch regression residuals (whole cohort) (*x*-axes). (**a**) Empirical values for all 15 items, (**b**) empirical values (Pr_exp and Rr_exp) and predictions of the effects of the scale distortion from the SPEs (Pr_theory and Rr_theory) (Equation (10), ∆δ=+2) primacy (Pr) and recency (Rr); PCA1 coefficients $a_{p,x}$ from Equation (6); distribution of a person’s ability (Equation (11)): ${\bar{\theta }}_{all}=-0.6\left(1.0\right)$ and σ=1.0.

**Table 1 entropy-24-00934-t001:** The RAVLT IR explanatory variables, empirical, CSE/quasi-theoretical, and theoretical task difficulty values and associated measurement uncertainties (*k* = 2) for each item.

	Explanatory Variables				Empirical		CSE Quasi-Theoretical	
	Primacy (Equation (A7))	Middle (Equation (A5))	Recency (Equation (A8))	Frequency (Equation (A6))	δ	Uδ	zR (Equation (3))	UzR
Item 1	0.00	6.31	−8.33	4.80	−0.81	0.28	−0.80	5.28
Item 2	−0.26	6.31	−7.38	4.85	−0.54	0.28	−0.24	4.86
Item 3	−0.66	6.31	−6.46	4.88	0.57	0.30	0.19	4.48
Item 4	−1.17	6.31	−5.58	3.99	1.38	0.36	0.34	4.16
Item 5	−1.77	6.31	−4.73	3.25	−0.64	0.28	0.42	3.92
Item 6	−2.43	6.31	−3.92	3.56	0.32	0.30	0.63	3.76
Item 7	−3.15	6.31	−3.15	4.31	1.16	0.34	0.84	3.71
Item 8	−3.92	6.31	−2.43	3.39	−0.09	0.28	0.66	3.79
Item 9	−4.73	6.31	−1.77	4.41	1.31	0.36	0.78	3.96
Item 10	−5.58	6.31	−1.17	4.06	0.64	0.32	0.55	4.22
Item 11	−6.46	6.31	−0.66	4.27	0.74	0.32	0.33	4.56
Item 12	−7.38	6.31	−0.26	5.44	−0.72	0.28	0.20	4.97
Item 13	−8.33	6.31	0.00	3.57	0.23	0.30	−0.66	5.40
Item 14	−9.30	6.31	0.00	3.24	−1.46	0.30	−1.38	5.87
Item 15	−10.30	6.31	0.00	4.93	−2.09	0.34	−1.86	6.36

## Data Availability

With regards to data availability, see: https://doi.org/10.5281/zenodo.5582001, accessed on 25 June 2022.

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
