# Peer review of "The Role of Entropy in Construct Specification Equations (CSE) to Improve the Validity of Memory Tests: Extension to Word Lists"

_entropy, 2022, doi:10.3390/e24070934_

Round 1
Reviewer 1 Report
The paper presents an analysis of the RAVLT within the Rasch model. While conventional analyses incorrectly treat the ordinal response data as if it is numeric and on an interval scale and conflate task difficulty and the ability of the test subject, the proposed methods handle the data appropriately and aim to isolate these conceptually-distinct sources of variance in the data.
This work is important for improving the quality of inference based on a very commonly used research and clinical instrument. A tremendous amount of psychological work involves the study of memory, and free-recall tasks are ubiquitous. The current work demonstrates the pitfalls of mishandling this data.
I do have critical feedback with regard to how this work is presented. Given the technical sophistication of the work, the writing should be as simple as possible. Some of the technical detail could be shifted into its own section, allowing the main paper to be more digestible for broad audience.
The writing felt redundant at times. Yes, the standard analysis approach to analyzing these data is wrong, but this point is hammered on again and again. It becomes distracting.
Given that this is a very concrete application---a memory task with 15 items---the paper spends a lot of time dealing in the abstract. I believe paper would be strengthened by keeping things simple and concrete at first, before building out the more general/abstract case.
Something that I struggled with: how can task difficulty differ between groups, yet still be separate from the subject's ability? The same task is administered to each subject---if performance differs between subjects within the same task, does this not reflect the subjects' ability? This was a conceptual road-block for me.
Finally, in the abstract, you write that you are extending your model from cases involving only syntax to a case that involves "semantics". While the items in this task are words rather than shapes, it's not clear to me that your proposed models relate to the semantics of the words. The only psycholinguistic variable considered is word frequency... more frequent words will tend to occur in more contexts and be associated with a larger set of other words. In other words, frequency is related to semantic factors. But, frequency is not semantics.
All in all, I found this to be a challenging read, but I believe it is important work that was conducted rigorously and reported thoroughly. The paper should be revised for clarity, simplicity, and concision.
Specific comment: It would be helpful if the graphs in Figure 2 had the same y-axis. Likewise for Figure 5.
Reviewer 2 Report
Please see the attached file

Round 2
Reviewer 1 Report
My concerns have been addressed. I have a small number of minor notes:
Please define RAVLT and CSE on first use in the paper .
Double check equation numbering in the main document (it seems that EQ2 should be EQ1).
It would be helpful to include person and item indexes in the Rasch equations to make it explicit that each person has their own value of theta, and each item has its own value of delta.
Reviewer 2 Report
I would like to thank the authors for taking into account my comments. The manuscript is much better in its present form.